# Overview of the Use of Probiotics in Poultry Production

**DOI:** 10.3390/ani11061620

**Published:** 2021-05-31

**Authors:** Katarzyna Krysiak, Damian Konkol, Mariusz Korczyński

**Affiliations:** Department of Animal Nutrition and Feed Management, Wroclaw University of Environmental and Life Sciences, Chełmońskiego 38C, 51-630 Wrocław, Poland; katarzynakrysiak0@gmail.com (K.K.); mariusz.korczynski@upwr.edu.pl (M.K.)

**Keywords:** poultry, probiotics, microorganisms, feed additives, microbiota

## Abstract

**Simple Summary:**

Probiotics are feed additives that have gained popularity in poultry production following the ban of antibiotic growth promoters (AGP). They are one of the more universal feed additives and can be easily combine with other additives. Probiotics, above all, have many advantages, including stimulation of the host microflora or immunomodulation. The statement “immunity comes from the intestines” has become more important in the poultry industry because probiotics have proven helpful in the fight against diseases of bacterial origin and against zoonoses. Positive effects on the organism have already been studied at the cellular level, where probiotics were responsible for changes in gene expression, leading to alleviation of heat stress. In addition to the health benefits, the utility value of the animals increases. The numerous advantages are overshadowed by a few drawbacks, which include the possibility of lowering semen quality in roosters and the diversity of production processes affecting the persistence of the probiotic. In addition to bird health, probiotics have improved the taste and quality of poultry products. Future prospects are promising as scientists are working to maximize the positive effects of probiotics by increasing the integrity of probiotics within the bird organism, taking into account, among others, bacterial metabolites.

**Abstract:**

In recent years, probiotics have become more popular in the world of dietary supplements and feed additives within the poultry industry, acting as antibiotic substitutes. Above all, probiotics are universal feed additives that can be used in conjunction with other additives to promote improved performance and health. Their positive effects can be observed directly in the gastrointestinal tract and indirectly in immunomodulation of the poultry immune system. Nutritional effects seen in flocks given probiotics include increased laying and egg quality, increased daily increments, and improved feed conversion ratio (FCR). There has also been an improvement in the quality of meat. This suggests producers can improve production results through the use of probiotics. In addition to these production effects, bird immunity is improved by allowing the organism to better protect itself against pathogens and stress. The lack of accuracy in the formulation of non-European preparations needs to be further developed due to unknown interactions between probiotic bacteria strains as well as their metabolites. The versatility of probiotics and the fact that the bacteria used in their production are an integral part of animal digestive tracts make them a safe feed additives. Despite restrictions from the European Union, probiotics have potential to improve production and health within the poultry industry and beyond. The following article will review the use of probiotics in poultry production.

## 1. Introduction

According to the Food and Agriculture Organization of the United Nations (FAO)/World Health Organization (WHO) definition, probiotics are “living microorganisms which when administered in adequate amount confer a health benefit on the host” [1]. For the product to be considered functional, the probiotic must have the following features: bacteria should be a component of the intestinal microflora, be resistant to the acid environment, easily adhere to the intestinal epithelium [2], and maintain the microflora present in the intestines at the appropriate physiological level. The poultry sector has strengthened its position in the agri-food industry over recent years. Currently, about five times more poultry are reared, and this has changed over 50 years [3]. Forecasts indicate that the size of the human population will be around 9.3 billion in 2050. It means that agricultural production and consumption will be 60% higher than today [4]. This will result in increased demand for meat, dairy products, eggs, fruits, and vegetables [5]. The consumption of vegetables and fruits is expected to increase by 35.8 g/person/day, red meat by 3.9 g/person/day [6]. Antibiotics can be used therapeutically and sub-therapeutically. Supporting antibiotic growth is a practice that has been known since about 50 years of the last century [7]. AGP are nothing more than subtherapeutic use of antibiotics. Some of the first ones used were, for example, streptomycin, tetracycline or avoparcin. They were supposed to cause an increase in body weight, or a decrease in the FCR [8]. The year 2006 was a landmark year for livestock production due to the EU ban on the use of antibiotic and hormonal growth promoters in livestock nutrition under Council Regulation (EC) No 2821/98.

The European Union’s strategy to ban AGPs has been adopted by countries such as Mexico, New Zealand, and South Korea. This is the responsible thing to do, although it may prove to be too radical a step for some countries. The USA, Australia, Japan, or Canada have enacted laws to partially ban antibiotic-derived additives and to exclude some [7,9].

Thirty probiotic preparations are currently registered in European Union. It is allowed to use preparations composed of several bacterial strains [10]. The most common types of microorganisms used to make probiotics are bacteria such as *Bifidobacterium* spp., *Lactococcus* spp., *Lactobacillus* spp., *Bacillus* spp., *Streptococcus* spp., as well as yeasts, such as *Candida* spp. [11]. The probiotic compositions of the present invention use isolates from probiotic bacteria that are designed to produce enzymes or substances that activate phytases, cellulase proteases or xylanases. Modern probiotics undergo a granulation process, which is a process that uses temperatures that are unfavorable to the bacteria. Therefore, for the production of these feed additives, for example *Bacillus* spp. producing spores. Thanks to heat-resistant spores, the probiotic does not lose its properties. This makes it possible to create feeds with added probiotic, which are also produced using a granulation process [12].

The most common method of administering probiotics on poultry farms is to add them to feed, while there are many other methods, such as gavages (vaccines or drops), sprays, granules, tablets, coated capsules, or sachets of powder. In addition to inflicting probiotics in the feed, growers are also opting to administer formulations in the water [12,13]. Each strategy has a different path to a common goal—the pathogen. The routes of action are explained in Figure 1.

One of the alternatives to the use of AGPs are probiotics. The introduction of probiotic bacterial strains improves immunity of the gastrointestinal tract, and consequently, the range of tolerance to adverse external stimuli. Probiotics are considered to be one of the more effective methods of microbial control and are not as detrimental to the environment as antibiotics [15]. Probiotics have many advantages and few disadvantages. The prospect of using probiotics in poultry production is clearly positive. Prophylactic use of probiotics occurs through antagonistic actions on other microorganisms and in competition for adhesion receptors or nutrients needed for their survival and some mechanisms like intestinal epithelial function and status. They also affect animal health as well as production performance, which will be developed later [16]. In 2015, the value of the probiotic market reached USD 33.19 billion. In 2020, the value of the market was USD 46.55 billion [11].

## 2. Antibiotic Growth Promoters

AGPs and Synthetic Growth Promoters (SGPs) are substances that had their heyday many years ago. In subtherapeutic concentrations they influenced the improvement of production indices such as body weight, FCR or daily gains. Their spectrum of action also included antimicrobial mechanisms, mainly targeting Gram-positive bacteria [17]. Their use to improve animal performance and rapid growth has maximized animal production results, while their mechanisms of action in this direction are not fully understood. Recent related knowledge highlights the possibility of manipulation of the gut microflora; AGPs have been shown to alter the diversity of gut bacteria, including beneficial LABs [18,19,20]. In the context to LAB bacteria, depending on the substance this effect varies [21]. Mechanisms of AGP action also reach to modulation of the animal immune system affecting its modulation; however, it has been shown that these reactions are different depending on the substance used, for example avilamycin affects the inhibition of bacterial protein synthesis, which release smaller amounts of proinflammatory compounds [22]. Moreover, the use of these feed additives has an effect on the amount of vitamins, nucleosides, amino acids, or fatty acids metabolized, interestingly, studies have shown an increase in their levels. In contrast, the most shocking information is the increase in polyunsaturated fatty acids (PUFA) [23]. Contradictory information on the topic of AGPs requires further study, and variation may originate from environmental differences affecting the study of external conditions, individual animal microflora composition, or animal health status. AGPs have been withdrawn due to undeniable residues in animal products, water, and soil, with negative consequences in terms of antibiotic resistance and allergies [24]. Their animal performance-enhancing and antimicrobial properties are undeniable, while their mechanisms of action need to be understood more and compared with the advantages and disadvantages of other alternative substances used in agriculture.

## 3. Other Feed Additives

### 3.1. Phytobiotics

Phytobiotics have been classified as plant-based feed additives that improve the health of farm animals [25]. Phytobiotics are characterized by the complexity of the biologically active ingredients [26]. There is great diversity in the structure and action of the active substances of phytobiotics, resulting in a variety of effects. Primarily, phytobiotics are used to replicate the effects of the banned AGPs including increased muscle mass, immunomodulation, prevention of diseases caused by microorganisms, and improvement the quality of animal products, they positively influence the taste and smell of meat and eggs [27]. Numerous studies have proven that plant compounds meet these criteria and provide the desired results. It makes them promising growth stimulants and immunomodulators [28]. Apart from the above-mentioned positive effects of phytobiotics, these compounds affect important nutritional aspects. The inclusion of phytobiotics in feed optimization shows an improvement in FCR, feed intake (FI), and average daily gains (ADG) [26,29].

In poultry production, phytobiotics seem to be promising and have good prospects for the future in the feed industry due to their natural origin [30]. However, the number of available compounds, as well as the variety of forms in which phytobiotics occur, is a disadvantage, as there is still a lot researchers do not know. According to Yang et al. [26], phytobiotics need to be studied in more depth because some of the plant-derived substances may contain toxic parts. Their efficacy is strongly dependent on the active ingredients concentration, their forms and current diet of animals. Therefore, before implementing a phytobiotic, it is necessary to perform tests on the compatibility of a given additive with the dietary model used to know the level of toxicity or the mechanism of the action itself [31]. Meta-analyses prove the heterogeneity of these compounds. Phytobiotics are synergistic, which results in numerous standardized preparations appearing on the market. For other additives, standardization is an easier procedure. In the case of herbal research, the same amount of the desired substance is not obtained every time. This depends on the part of the plant from which the substance comes (root, stem, leaf, flower), the type of soil, the agrotechnical treatments applied, and the climatic conditions. The effects of both phytobiotics and probiotics are similar. Better effects are infrequently demonstrated for animals fed with probiotics [32]. Both groups of feed additives improve production results, animal health, and the quality of the final product. In modern feed science, the combinations of additives are known. Probiotics used together with other additives bring better results and therefore constitute a universal base of feed additives. The studies have shown that the combination of a phytobiotic preparation with a probiotic has beneficial effects on blood parameters [33] and has a strong antimicrobial effect, as demonstrated by the research on *Escherichia coli* [34].

Probiotics have been used for a long time in the feed industry. They are a proven additive that can be easily combined with another (usually a prebiotic) to form an effective synbiotic. The synergistic properties of phytobiotics, on the other hand, are limited to the group of plant compounds, and as a result, their combination with other feed additives is less common than that of probiotics. It is the extensive knowledge, versatility, and technological progress in the production of probiotics that makes them a leader on the feed additive market.

### 3.2. Surfactants

Surfactants are substances consisting of molecules composed of two parts—hydrophilic and hydrophobic. This first group is strongly polar. Surfactants can be divided into 4 groups: non-ionic, anionic, cationic, and amphoteric. They differ primarily in the purpose of use and the environment in which they will be used. They are widely used for both humans and animals [35]. Similarly to the above-mentioned feed additives, surfactants undoubtedly show positive effects on poultry production and health. They have a bactericidal and bacteriostatic effect. Surfactants can be used alone and in combination with other additives. For example, hydrogen peroxide is used to reduce the amount of *Salmonella* spp. on the surface of the eggs. The presence of a surfactant improves its efficacy, contributing to safe consumption of eggs [36]. According to Keener et al. [36], the use of surfactants at slaughter or post-slaughter may radically reduce the risk of campylobacteriosis, salmonellosis, and coliobacteriosis, which are among the leading zoonoses. The decrease in the number of microorganisms causing the above-mentioned zoonoses helps in developing control strategies [37]. The presence of surfactants increases the activity of digestive enzymes, such as xylanase. For example, Safety Data Sheet (SDS) has improved the activity of xylanase by a factor of 1.29 [38]. Surfactants are also used in the production of hydrolytic enzymes, which affect the secretion of certain proteins, and in turn, affect nutritional effects [34]. Some feed grains, such as oats, barley, and wheat, are distinguished by the presence of non-starch polysaccharides (e.g., glucans), with undesirable properties, such as viscosity. Their effect can be compared to the gums that clog the light of the intestinal mucosa. In this case, the use of a surfactant, such as saponins, may prevent this type of occurrence [39].

Synthetic surfactants are used in many industries; however, due to contaminated environments, biosurfactants are increasingly used. Biosurfactants are surfactants of biological origin, characterized by low toxicity and high biodegradability. The probiotics discussed in this article may form part of surfactants called microbiological biosurfactants. In addition to their antiparasitic and antifungal effects, surfactants still have a limited field of action in veterinary medicine. The addition of a probiotic to surfactants makes it more integral with the animal organism through bacteria that already exist in the digestive tract of birds. Compared to biosurfactants, these probiotic additives are used in clinical and therapeutic settings, significantly increasing the efficiency of these feed additives [40]. Synthetic surfactants, including SDS, qualify as a group of detergents. Their use does not have an inert influence on the environment, leaving negative traces on aquatic and terrestrial environments, causing pollution [41].

Therefore, as more manufacturers and consumers increase their awareness, the use of biosurfactants increases. With the improvement of surfactant formulas, probiotics once again prove their versatility and compatibility with other feed additives.

### 3.3. Organic Acids

Natural methods of fighting against pathogenic microorganisms existing in birds digestive tract is the use of organic acids. For example, acidity in the stomach minimizes the risk of developing diseases caused by bacteria. Organic acids work on a similar principle, supporting a reduction in the pH of the intracellular pathogenic bacterium. This leads to inactivation of enzymes and complete destruction of the bacterial cell [42]. Like the majority of feed additives, the popularization of organic acids began with the ban on AGPs. Admittedly, antibiotics show minimally greater effects on DWG, FI, and FCR, although the reasons for the ban of antibiotics are clear [43,44]. The antimicrobial effect functions by destroying the structure of the wall and cell membrane and preventing further multiplication of genetic material. However, it is important to remember the specific spectrum of action of a given acid, therefore, despite the versatility of its functions, its range of action is narrow. These feed additives can result in reduced amounts of *Salmonella* spp. in the carcass itself and the light of the small intestine. Adding acids to feed and water increases the hygiene of the breeding process by generally reducing pathogens [45,46]. The use of acidifiers also increases the availability of nutrients, while reducing undesirable metabolites of pathogenic microorganisms. Studies show that the number of cup cells and the height and width of intestinal villi increased when citric acid was combined with acetic acid, and contributed to the above-mentioned positive effects [47].

The use of organic acids as feed additives was expected to improve production results. They are often added together with probiotics as one preparation. Weight gain occurs as expected during administration. In flocks where a decrease in egg weight was observed, it was minimized. Moreover, the organic acids themselves increase the T3 hormone concentration, resulting in increased metabolism [48]. Moreover, favorable results occur in the studied lipidogram, where the cholesterol content in blood is reduced [43].

Despite positive results with multiple parameters, the use of organic acids still raises doubts. First, the rapid acid metabolism in the small intestine is an important limitation [1,48]. This translates into doubtful presence of acids in the lower gastrointestinal tract. The effectiveness of organic acids is limited by the condition of bacterial microflora, the composition of the acid product, and the current state of health of birds. Additionally, the initial phases of administration of preparations may cause negative effects on lactic acid bacteria (LAB) already present in the digestive tract. If the bacterial flora of the intestines is in a bad condition, the acids may be counterproductive [49]. Other disadvantages of organic acids are their instability, unpleasant smell, and corrosivity. The manufacturer’s solution for the effective use of acids as a feed additive is to administer them in the form of salts, microcapsules, and buffered or encapsulated substances [50].

The above analysis shows, to a greater extent, considerable versatility of probiotics as feed additives. Their use is less demanding than that of other preparations, where growers would need to focus their attention on factors that significantly affect their effectiveness.

### 3.4. Chitin

The use of insects in nutrition and feed is becoming increasingly popular. Chitin is one of the increasingly popular bioactive feed additives. It is a polymer that is included in the structures building the carapaces of insects, fungi and crustaceans (N-acetyl-D-glucosamine—GlcNAc). This is a component that is difficult for some animal species to digest. The enzyme acidic chitinase (Chia) is found in higher amounts in omnivorous animals. Thanks to this hydrolytic enzyme, chickens can break down chitin [51,52]. Typically, chitin is fed in feed in the form of meal from the exoskeletons of insects, such as Black soldier fly (*Hermetia illucens*), crickets, or marine animals such as shrimps [51]. It is one of the ways of using shellfish waste, which makes it more ecological, according to Hossain et al. [53] it is a component of “green technology” in animal nutrition. Chitosan is a derivative of chitin after partial deacetylation. Among other things, it influences the morphological structure of the intestines; it was noticed that the villi of the jejunum elongated and the depth of the intestinal crypts decreased. Chitin, on the other hand, proved to have a positive effect on carcass performance and internal organ traits more than its deacetylated form. Both performance and carcass quality can be improved by reducing triglycerides in the liver and pectoral muscle, among others [52]. Chitin-fed chicken groups had the best FCR while consuming the least amount of feed [54,55]. There are also known antimicrobial properties of chitin, which has antifungal [56,57], antiviral [58,59], and antibacterial [60] effects. The intestinal microflora is also enriched when this component is fed to chickens, as it affects the quantitative increase of LAB and the decrease of pathogenic bacteria [61,62]. The feed additive causes a numerical increase in the expression of genes (PepT1, EAAT3, and SGLT1) responsible for nutrient transport in the intestine, while the mechanism is not yet fully understood [60]. Another beneficial aspect is the reduction of ammonia in bird excreta and a significant increase in butyric acid [63].

While its presence in feed provides many benefits and advantages, inconsistencies have been noted in scientific research. Namely, chitin can cause a decrease in nutrients digestibility. Khempaka et al. [63] suggested that excessively high levels of this feed additive may contribute to reduced digestibility and growth performance in chickens, and the conclusions of their study showed the most beneficial level of chitin to be below 2.8% in the feed.

The analysis showed many advantages of adding chitin or chitosan to poultry feed. Considering the antimicrobial health properties, the aspect of using the waste as a feed component is an ecological solution. Dosage and integrity with other feed components can have the desired effect, while growers need to carefully consider the amount of this component in the feed to avoid the opposite effect.

### 3.5. Medium and Long Chain Fatty Acids

Fatty acids have been known for a very long time, mainly for their antimicrobial properties used externally in the form of soaps. However, internal use in both humans and animals has many benefits. Medium chain fatty acids (MCFAs) are popular for their antiseptic properties [64]. These are acids with 6–12 carbon atoms, while commercial products typically use acids with 8–10 carbon atoms in their structure. Their main sources are cow’s milk and coconut oil [65], while other sources are breast milk and palm kernel oil. Examples of MCFAs are caproic acid, caprylic acid, capric acid and lauric acid [66]. They possess coccidiostatic [67], antibacterial, and antifungal [68] properties due to their effect on the structure of microorganisms [64]. This is due to changes in the physicochemical properties (due to the anionic part of the fatty acids) of the bacterial habitat and the effect on bacterial gene expression. The exact mechanism is not yet understood [65].

The synergism of MCFAs with other feed additives makes them effective nutritional supplements to support productivity. Both organic acids and MCFAs cause an increase in body weight and a decrease in animal mortality by improving the environment for gut microbes [69,70]. Dietary supplementation with MCFAs and *Moringa oleifera* leaf meal (MOL) resulted in a marked decrease in FCR, higher live body weight as well as body weight gain. Crude protein conversion was also improved and the growth rate was higher compared to the control group. Another important aspect is the improvement of the blood composition, namely an increase in the number of white blood cells, lymphocytes, and a lower concentration of heterophils. The supplements had an indifferent effect on intestinal microorganisms—an increase in the amount of *Lactobacillus* and a significant decrease of *E. coli* [71]. Mention should be made of the effect of the acids themselves on blood lipid profile with MCFAs supplementation. They resulted in a decrease in blood total cholesterol, LDL cholesterol, and glucose with a concomitant increase in HDL cholesterol [72].

Long chain fatty acids differ (LCFAs) from MCFAs in structure, their chains contain a minimum of 12 carbon atoms. The different structure of acids manifests itself in differences in many characteristics and processes, such as solubility, water content, absorption, and fat transport. LCFAs as chylomicrons can be transported through the lymphatic system, while they have lower oxygen stability compared to MCFAs [65]. There is also a visible difference in the energy load of fatty acids, which affects the efficiency and energy balance of animals. An important family of LCFAs is the PUFA. They are linked to many health benefits. Microalgae (MA) and fish oils are rich in these acids (docosahexaenoic acid (DHA) and eicosapentaenoic acid (EPA)). Supplementation with both algae and fish oils has been shown to increase the amount of these beneficial fatty acids in meat and eggs. Unfortunately, this is associated with unfavorable sensory impressions of zoonotic products [73].

Omega-3 and omega-6 fatty acids play an extremely important role in the development of chickens at the embryonic level, especially during the development of the immune and nervous systems. A holistic approach to the diet of breeding hens with efficiently controlled diets will increase hatchability rates, viability as well as improve embryo health [74,75]. Omega-3 fatty acids have anti-inflammatory properties due to decreasing cytokines. Their beneficial effects are seen in the increase of bone strength and improvement of their mineral composition. Unfortunately, omega-6 fatty acids have been linked to higher rates of depression and heart disease. Even so, their large range of health improvement is seen in reducing coronary heart disease and decreasing LDL cholesterol levels. Both types of omega acids have a positive effect on semen quality and density. The use of marine algae has a beneficial effect on the n-6/n-3 PUFA ratio [76]. Unfortunately, an undeniable disadvantage is the high oxidation of LCFAs, which contributes to negative consumer perception. Future research should focus on using them while taking care of the sensory experience [77].

## 4. Probiotics as Immune Helpers

Both antibiotics and probiotics provide antimicrobial substances (with modification of intestinal pH and in combination with glucose) at a similar level of efficacy as organic acids, bacteriocin, short-chain fatty acids (SCFA), or hydrogen peroxide [78]. The study on popular *Bacillus* spp., in particular, showed immunomodulatory effects; the expression of TJ protein adhesion molecule (zonulin 1 and occludin) increased. The result is an increased efficiency and integrity of the intestinal barrier. Probiotic microorganisms have the ability to balance proinflammatory cytokines while increasing the amount of anti-inflammatories, including IL-10 and TGF-β [79]. The administration of these feed additives has a positive effect on the level of immunoglobulins M and A. The percentage of total antioxidant capacity (TOAC) in serum has also increased [80]. Moreover, reports indicate that *Lactobacillus rhamnosus* has the ability to activate the receptor responsible for epidermal growth in the intestine. This results in a reduction in intestinal epithelial apoptosis, which is an important component in the fight against gastrointestinal diseases [81]. The poultry digestive tract has an impressive number of microorganisms, commonly referred to as microbiota. The number of bacteria in the gastrointestinal tract is estimated to be between 10^10^ and 10^11^ CFU/g of intestinal content. The most common bacteria are *Lactobacillus* spp., *Bifidobacterium* spp., *Ruminococcus* spp., *Clostridium* spp., and *Bacteroides* spp. The functions of these microorganisms can be generalized to preserve homeostasis in the body. The individual functions of intestinal bacteria are very different. Their role is, among other things, to increase the energy efficiency of feed by fermentation. The products of which are SCFA or the breakdown of indigestible nutrients, such as polysaccharides for monosaccharides. It is estimated that 10% of the energy from feed is derived from intestinal bacteria. This gives a better energetic use of the feed, and the assimilation of important nutrients is easier.

Microbiota is an inherent segment of the intestinal ecosystem and has been given the title of a metabolic organ that adapts to the physiology of the host organism. Bacteria have an influence on the very structure of the intestine and its functioning—intestinal microorganisms enlarge the villi and intestinal crypts. The intestinal microbiome can affect its morphology, particularly regulating the immune processes that occur. Figure 2 compares the effects of a probiotic as well as an antibiotic with a standard diet. Accompanied by antibiotics, the intestinal mucosa was described as damaged with increased defects in the tips of the intestinal villi and changes in the intestinal mucus layer. On the other hand, the administration of the probiotic gave the opposite effect, namely, a diet enriched with a probiotic preparation caused the development of the intestines [11].

The most important component of the microflora is a gene reservoir that encodes the enzymes necessary for metabolic changes. Poultry do not have polysaccharide lyase genes or glycosidic hydrolysis, which are essential for the distribution of polysaccharides. Therefore, the presence of bacteria enables and facilitates this process [82]. The metabolism of bacterial microflora in the use of probiotics is crucial. It is unique for each individual. Probiotics lead to variability in its composition, which helps in eliminating pathogens. The composition of the microbiome is also influenced by other factors such as the quantity and quality of nutrients or the composition and balance of the feed itself [83] This slight modification usually involves an increase in the number of *Bifidobacterium* spp. and *Lactobacillus* spp. bacteria, which are the most numerous in the intestinal microbiome composition. Other effects include decreased activity of bacterial enzymes and decreased stool pH [84]. The mechanism of action is based on the interaction of probiotic microorganisms with mucosal epithelial cells, thus inducing specific CD-206 and toll-like receptor (TLR)-2 cells [85]. In the study, the decrease in stool pH and intestinal environment was caused by increased concentrations of acetic acid, lactic acid, and volatile fatty acids (VFA). The acidified environment is conducive to the development of intestinal microorganisms, supporting the fight against pathogenic microorganisms, and supporting the organism’s natural defense mechanisms [11].

## 5. Advantages

Probiotics are preparations that have a positive effect on the gastrointestinal tract and immune system, primarily due to filling in gaps that antibiotics leave behind. These include sterilization of the animal’s organism from its natural defensive barrier and the abuse of antibiotics, which causes increased resistance to their effects, and consequently, increased doses or use of more aggressive substances. There are many possible modes of probiotic actions. As a result, probiotics have more room to act against pathogens, which has been set at Figure 2.

Probiotics typically used in the poultry industry are strains of the genus *Bacillus* spp. Additives based on these bacteria are known for their higher tolerance to high temperatures and acidic pH. The advantage of using these bacteria is their increased quality and widespread use in probiotics, making them more versatile in use. Studies indicate that the probiotic often reaches the intestines intact due to low stomach pH or elevated body temperature of the bird [87].

The statement “immunity comes from the intestines” has become more significant in the poultry industry with the emergence of probiotics. The use of probiotics against diseases of bacterial origin is more common after the ban on AGP. This is largely due to previously knowledge about bacterial interaction, in which the microorganisms compete with each other for mechanisms and substrates to survive [87]. The supply of probiotics affects the predominance of probiotic bacteria, reducing the number of pathogenic microorganisms.

Probiotics have already been identified as an alternative to AGP. Given the diversity between strains and species of microorganisms, a particular feature is that they do not leave residues in animal products, as opposed to antibiotics [88]. The use of probiotics may actually improve the quality of meat and eggs, which is beneficial as the demand for meat and eggs continues to grow as the earth’s population increases. Providing healthy and safe products is the main goal of poultry farming. Probiotics have become alternative growth promoters, increasing product quality and productivity of animals [89].

It is known that probiotics are not indifferent to the internal organs of the bird. An increase in the weight of certain internal organs may be noted. The authors of the study assessed the effect of probiotics on the mass of some organs as significant (spleen, thymus) [90,91]. The gut is one of the most important parts to evaluate the effectiveness of a probiotic. Improving the quality—length, density of the intestinal villi, as well as increasing the crypts is one of the most desirable effects that improve nutrient absorption and allow proper colonization of bacteria. Apart from the mentioned aspects, an increase in the mass of some gastrointestinal tract sections including the cecum [92], including an increase in the thickness of the mucosa was observed [93]. In addition to the mass, the length of the bowel also increased significantly [94]. Peyer’s tufts, which take part in the body’s immune response, located in the mucosa and submucosa of the small intestine also increased in number, which may be an important aspect in the fight against the pathogen [95,96], but the thickness of the mucosa itself and the intestinal musculature did not change their characteristics with the use of the probiotic [97]. A diet supplemented with probiotics resulted in bone strengthening by increasing calcium and phosphorus retention. This was manifested by improved bone mineralization as well as increased serum concentrations of these elements [98]. It also positively affected wall thickness (medial and lateral), ash percentage and tibial index, size and mass of tibia, and femur as their density [97]. Probiotics also allow to prevent and reduce the effects of pathologies and diseases of the bone system, including bone resorption. C-terminal telopeptide of type I collagen is an indicator of the level of disease management, supplementation with these feed additives can prevent bone resorption [99]. Broilers fed with a probiotic immediately after hatching have a lower frequency of lameness, and in poultry suffering from bacterial chondronecrosis, there is an improvement in walking [100]. Chicks also had larger body and bone dimensions than chickens who did not have a probiotic in their diet [93].

Probiotics affect not only the physical properties of meat, but also the chemical properties. It depends on the composition and concentration of a given probiotic. Probiotics have a positive effect on overall carcass weight, with abdominal fat reduced, leading to improved poultry carcass quality. Increased carcass yield was noted in chickens, regardless of sex [101]. This represents an important economic aspect. The presence of probiotic feed additives results in increased absorption of nutrients, including amino acids needed to build tissues resulting in increased carcass weight [101,102]. The protein content of thigh meat and breast meat has improved. Depending on the concentration of probiotic administered, the effect of the experiment changed, the concentration of 0. 160 g probiotic/liter of drinking water caused an increase in water absorption in pectoral and femoral muscles, while the concentration of 0.175 g probiotic /liter of drinking water gave the opposite effect [100]. According to Duskaev [100], probiotics also have a positive effect on increasing the amount of chemical elements in the liver (Ca, K, Mg, Mn, Si, and Zn) and chicken breast muscles (Ca, Na, Co, Cu, Fe, Mn, Ni, and Zn). Researchers have also shown effects on the microstructure of meat. Meat homogenization and probiotic supplementation decreased myofibril destruction in pectoral muscles [89]. Probiotics also have a moderate effect on cohesiveness, firmness, chewiness and elasticity of cooked breast meat [103].

The probiotic containing *Bacillus licheniformis* has resulted in a significant improvement in the amount of protein and in the improvement of essential amino acids [103]. The use of cholesterol by probiotic bacteria contributed to the reduction of cholesterol in meat [104,105].

The administration of probiotic microorganisms improves the profile of fatty acids by reducing their saturation. It is quite rare, but applying *Aspergillus awamori*, *Saccharomyces cerevisiae* or a combination of these promotes this effect. The TBARS tests have proven that probiotics increase the oxidative stability of meat [106,107]. An important aspect for the consumer is to improve taste, smell and color of poultry meat [108]. Studies have also shown a reduction in the number of pathogens on in vitro models [109,110]. The same additives were tested on in vivo models and the results were equally satisfactory. When broilers and turkeys were examined, the occurrence of *Salmonella enteritidis* and *Salmonella typhimurium* in neonatal broilers decreased after just one hour. This was probably due to innate immune stimulation or bacterial (competitive) interactions. This may include competition for SCFA, receptor sites or bacteriocin production [87].

Positive aspects of probiotic administration can also be distinguished in laying poultry. The first visible consequence of probiotic supplementation is the improvement of laying. While the control group’s laying was falling, the test group was performing better. Moreover, none of the hens from the study group laid less than a 48 g size egg, the majority of them were 73 g to above 63 g size eggs [111]. The quality of the eggs was improved by increased strength and shell thickness [112]. In flocks where probiotic was administered, fertility and hatching capacity of eggs improved [113]. In the groups of laying hens with probiotic, egg dry matter and percentage of protein increased. This can cause an increase in the weight of eggs [114]. The improvement of the quality of life of animals due to the probiotic also has positive economic effects, as the amount of broken eggs is reduced due to a stronger shell structure.

In addition to animal productivity, it is important to minimize the damage that intensive animal production causes to the environment. It appears that probiotics reduce the amount of nitrogen and phosphorus excreted to a large extent [115]. Increased population of probiotic bacteria improves the activity of microbial enzymes, thus improving digestibility and absorption of nutrients. These useful microorganisms modulate the biosynthesis and degradation of mucin affecting intestinal function. The result is better absorption of nutrients. Probiotic microorganisms are also responsible for the induction of the breakdown of proteins into nitrogen, thus improving the use of protein and nitrogen. The consequence of better use of nitrogen is a reduced amount in the feces. In the case of phosphorus, microbial phytase is helpful. Studies on laying hens have shown that the administration of phytase can reduce phosphorus excretion by up to 47% [116]. They are used primarily to minimize the unhealthy effects of phytates. A common phenomenon in the poultry diet is the administration of probiotics and phytase. In order to minimize the administered substances, an improved probiotic based on *Lactococcus* or *Lactobacillus* was developed. The phytase maple AppA was taken from *Escherichia coli*. Experience has shown 99% similarity of both sequences. The introduction of modified bacteria into broiler diets revealed a positive effect, namely that the amount of phosphorus in the feces has decreased. The phytase output of both probiotic and commercial phytase did not differ significantly, while the use of an improved probiotic makes it possible to dispense with the additional purchase of commercial phytase [117,118]. Table 1. presents single and multi-strain commercial probiotic preparations which can used in poultry production. These are some of the most popular bacterial strains used to make probiotic supplements.

## 6. Disadvantages

Probiotics have many undeniable advantages. Common use of *Lactobacillus* spp. has resulted in more research on the topic. Research shows the properties of the probiotics, including those that may disqualify them from use in the feed industry. An overdose of probiotic has less negative effects than a deficiency of probiotic. However, when using probiotics on a reproductive flock of roosters, the dose administered should be carefully determined. Research has shown that an overdose of this additive may cause a deterioration in the quality of the semen. The effect of the concentration of *Lactobacillus* spp. within the cloaca has a direct effect on the semen. The addition of probiotics with *Lactobacillus* spp. could be one of the causes of infertility in a herd of rooters. *Lactobacillus* spp. is naturally present in the semen, but prolonged administration of the probiotic may result in high concentration within the cloaca, leading to a decrease in semen quality [120,121].

When providing probiotic agents, appropriate research on the method of storage should be performed. The type of dehydrator used influences the structure of fatty acids and secondary protein structures of bacteria. In one study, a probiotic containing *Lactobacillus acidophilus* and *Lactococcus lactis* ssp. was tested. Changes in the structure of fatty acids (FA) and secondary bacterial structures were observed when higher than normal room temperatures occurred. In the above experiment probiotics were stored in plastic bags with various dryers. Of the NaOH, LiOH, and silicone granules, it was sodium hydroxide that contributed to the maintenance of the respective structures. Proper storage and development of packaging technology is extremely important to retain the right product properties in order for it to work properly [122].

Other research results also show the opposite side—no effect on individual organs of slaughter chickens. In the study, the primary parts of the carcass considered were the heart, liver, spleen, thighs, breast, back, and neck, whose weight was not significantly increased. The pH also did not change significantly [101]. Differentiation of the effects of probiotics supplementation also concerns the publications investigating the influence of these feed additives, or rather the lack of it, on meat quality [123], structure [124], or carcass quality [125]. According to Behrouz et al. [126], there are many contradictions in the research on the effectiveness of probiotics and their actual effects on avian body and production performance. It emphasizes the dependence of effectiveness on the dose and type of microorganisms and the conditions of administration of these dietary supplements. Table 2. shows the variation in the performance of the different feed additives in terms of efficacy between formulations as well as the effectiveness of the formulation with respect to the different organs.

The concept of administering probiotics seems reasonable due to the lack of negative effects of administration. On the other hand, it is a difficult task to introduce optimal strains under optimal conditions due to the possibility of their selection and the possibility of their combination with each other. There are many interspecies combinations and strategies that influence the effectiveness of the administered formulation. The European market is dominated by single-species preparations, while non-European markets use compound cultures of undetermined composition or multispecies preparations [127,128]. Therefore, the phenomenon of synergism between different bacterial strains is not sufficiently understood [129]. The efficacy of probiotics depends on the dose/day, probiotic strain, condition, and types of microorganisms residing in the gut [16]. The environmental problem causing discrepancies in research results may be the increasing air temperature and the occurrence of heat stress, which is becoming an increasing problem among poultry farmers [130]. Both different breeding, nutritional aspects, water quality or stressors affect the experimental results [131]. Feed structure and density have been proven to affect probiotic exposure [132]. Meta-analyses have also shown that the number of animals matters, namely, exercising control over hygienic conditions is more difficult with a large number of animals used in the experiment, which may affect the effectiveness of the probiotic bacteria. The state of the microflora is also important in the early stages of the host’s life, as the probiotic bacteria supplied can modulate gene expression in epithelial cells, thus creating a favorable environment for themselves to live in, which is important in later stages of life for the composition of a sustainable gut microflora [133]. An important aspect in choosing a probiotic preparation and its effectiveness is to consider the interaction between the different strains of probiotic bacteria in the feed additive and between the gastrointestinal microflora [134]. Namely, genomic diversity, which is related to the metabolic processes of bacteria with specific microbial functions in the gut, may be important. Observation of industrial strains allows their protection and the discovery of particular interactions between bacterial populations [135].

## 7. Zoonoses

The overriding goal of maintaining the food industry and its economy is food safety. A dangerous and widespread zoonosis is salmonellosis. In Europe, 82,694 cases were reported in 2013 [136]. According to the Rapid Alert System for Food and Feed (RASFF) data, the number of notifications about the threats of microorganisms entering this unit has been growing since 2006. RASFF is the body that verifies and then submits the relevant data to European Food Safety Authority (EFSA) [137]. Data from EFSA shows that *Salmonella Enteritidis* is responsible for 39.5% of human salmonellosis cases [138]. Approximately 29% of the cases of salmonellosis that people suffer from come from the poultry industry [138]. Probiotics, by means of competitive exclusions, prevents colonization of this microorganism. The contribution of probiotics to the diet undeniably controls the presence of the microorganisms *Salmonella* spp. A significant difference was observed between a control group with no probiotic and a study group where a probiotic based on *Bacillus* spp. was administered orally. The number of bacteria in the laying hen caecum had been reduced by 33% [139]. However, combinations of probiotics (i.e., those containing more than one strain of probiotic bacteria) give better results. The combination of *Lactobacillus salivarius* 59 and *Enterococcus faecium* PXN33 reduced colonization of the pathogenic micro-organism in the caecum, ileum, and colon compared to trials where probiotics were applied individually. The study suggests that inhibition of the spread of harmful bacteria was caused by a decrease in pH induced by probiotic bacteria [140].

*Bacillus subtilis* PY79hr is a solution for the emergence of salmonellosis, but is mainly known for controlling coliobacteriosis, which, like salmonellosis, is one of the most serious zoonoses in the poultry sector [141]. In 2013, 6043 cases of coliobacteriosis were observed in Europe [136]. The appearance of *Escherichia coli* in the flock can deform the digestive system of birds. The efficacy of probiotics in other zoonoses has been confirmed. In regards to coliobacteriosis, positive effects of the probiotic are observed with normalization of intestinal microflora and reduction of intestinal dysbiosis. A combination of *Lactobacillus casei* 1.2435, *Lactobacillus rhamnosus* 621, and *Lactobacillus rhamnosus* A4 have been shown to improve animal health [142]. Further evidence of the effects of probiotics on coliobacteriosis was evident in the reduction of these bacteria in broiler feces. The colonization of *E. coli* decreased in direct proportion to the increase of *Enterococcus* sp. This research highlights the antimicrobial activity of this probiotic strain [143].

Campylobacteriosis is the most common cause of bacterial gastroenteritis in the world [14]. Campylobacteriosis and salmonellosis are leaders in foodborne diseases. The increase in the incidence of campylobacteriosis is about 2–3 times higher than the number of salmonellosis cases [144]. *Campylobacter jejuni* is a thermotolerant bacterium, which is the most common source of human nutrition diseases, originating from broilers. Of the reported cases, 10% were reported to be hospitalized [145]. Approximately 0.2% of them were fatal [146]. In 2013, campylobacteriosis was the most notable zoonosis in the EU [147]. A small part of the data is available to summarize the costs of Campylobacteriosis diseases and their consequences, but with the available data, the costs are estimated to be about 2.4 billion euro per year [148]. The first step to address this issue is to prevent the spread of pathogens and then to control them at farm level. It must be a quick reaction with the choice of the right preventive strategy for the herd. There are ways to fight this disease. Table 3. shows possible strategies to combat campylobacteriosis. There are few effective solutions. Campylobacteriosis differs from salmonellosis and coliobacteriosis because of its increased difficulty to counteract. Consistent immune interventions are important as there are many different *Campylobacter* strains, there is inability to vaccinate in order to produce a strong and sustained immune response [149].

One of the greatest advantages of probiotics is their versatility, which manifests itself in three ways. They enrich the intestinal microflora, support the immune system, and act directly on the pathogen.

## 8. Heat Stress

Heat stress has become a serious, global problem. It is caused by excessive high temperatures in the environment in which the animals are kept. Causes of heat stress may include climate change (in the form of higher than average summer temperatures), faulty ventilation, poor housing construction, or excessive stocking density. Heat stress creates problems for the poultry industry, having negative impacts on the immunology, physiology, and microbiology of birds. Excessive ambient temperature results in disturbance to intestinal morphology, disrupting absorption and digestion, making the penetration of toxins and luminescent antigens into the blood easier. Ischemia and hypoxia of intestinal tissues is another negative effect of heat stress. These disorders are caused by the redirection of systemic blood flow from internal organs to peripheral circulation [150]. Birds experiencing heat stress are hypersensitive to corticosterone [151], which can delay the proliferation of intestinal cells, which in turn leads to a decrease in the height of intestinal villi and a decrease in the depth of intestinal crypts [152]. In addition, corticosterone is an activator of pro-inflammatory reactions in the intestine [153]. This is the cause of the aforementioned digestive and absorption disorders. Climatic change causes a decrease in animal productivity and disruption of homeostasis, which also includes changes at the molecular level. The expression of genes, such as heat shock proteins (HSP), is likely to change. This includes HSP 40 and HSP 90, which are active in self-regulation and compensation to maintain homeostasis [130]. In response to heat stress, an adequate immune response occurs, but its harmful effects cannot be excluded. The exact process of harm is presented in Figure 3.

In the presence of heat stress, probiotics started to be used in poultry production, where the emphasis was on production capacity. Feed additives containing *L. pentosus* ITA23 and *L. acidophilus* ITA44 had a positive effect on growth and feed conversion factor (FCR) [154]. An important nutritional aspect that improved was the ratio of feed consumption to the broiler growths obtained [150]. Moreover, in laying flocks, the administration of probiotic drugs has resulted in increased laying [155].

There are many disputable hypotheses as to why probiotics eliminate problems caused by heat stress. The thesis supported by the literature is that probiotics increased the activity of thyroid hormones, whose secretion was abnormally reduced during heat stress. Thyroid hormones have a significant impact on the metabolism of the body and normal growth and development. Returning T3 (triiodothyronine) and T4 (thyroxine) hormones to the correct level could reduce the number of abnormal changes in the intestinal tissues and increase the growth of birds [156,157].

In economic terms, the negative effects of heat stress may include reduced feed consumption, reduced daily growth, and poorer egg production. In the humid tropics, highly productive poultry have higher feed consumption and heat production due to excessive metabolic activity. Abnormal and reduced animal performance should be eliminated by developing appropriate feeding strategies [130]. In an effort to address these issues, probiotics have gained popularity as feed additives. Within the literature, there are disputable opinions on the effectiveness of probiotics on microbial agents in poultry, which are disharmonized during heat stress [130]. The benefits of using probiotics, as described previously in this article, prevail.

Probiotics were repeatedly tested under changed temperature conditions and with a developed dietary program. In the group of broilers with increased air temperature up to 35 °C, a positive effect on the final body weight was observed. Increased expression of genes responsible for the transport of sugars was also observed, indicating an improvement in its absorption [154]. Probiotic nutrition can even contribute to increasing the weight of the thoracic muscle and reducing HS effects, such as reducing the water content of broiler carcasses. Increased air temperature causes a decrease in pH, which leads to denaturation and impairment of protein function. Both immediately after slaughter and at retail sale, the water loss in the pectoral muscle of hens given probiotics was lower [158].

Antioxidants are increasingly being combined with probiotics, one of them being selenium. In animals with high production capacity, the heart muscle is particularly at risk because of the imbalance between the muscles that require large amounts of oxygen and the organs that supply it, such as the lungs and heart [159]. Selenium is one of the trace elements that plays a key role in the body, including protection of red blood cells against the harmful effects of free radicals, and is a component of a strong antioxidant, glutathione peroxidase. In summary, its presence is necessary for the formation of normal enzymatic systems such as superoxide dismutase and glutathione peroxidase [160]. Selenium supplemented diets with the addition of a probiotic showed the best effectiveness in reducing heat shock proteins. Providing selenium, *L. acidophilus*, and *S. cerevisiae* separately was not as effective [161].

The effective use of probiotics in eliminating heat stress is also confirmed by studies conducted on laying hens. The administration of probiotic to animals exposed to heat stress had a positive effect on the performance indicators of hens, such as average daily feed intake and egg weight, which was somehow higher. This was due to the increased thickness of the shell, its strength, and the amount of protein in the eggs. The administration of probiotic bacteria resulted in the improvement of intestinal microflora by reducing pathogenic microorganisms. The intestines also had improved integrity [162]. In another experiment on laying hens, probiotic reduced the amount of reactive oxygen species in the ileum and the caecum. Probiotics have also been shown to reduce the serum concentration of dimalone aldehyde, which is a product of reactions that occur in the body during oxidative stress [114].

## 9. Future Properties of Probiotics in Poultry

European Union legislation has imposed huge changes in animal nutrition standards. Increased consumer awareness has also forced poultry producers to eliminate conventional anti-microbial therapies. As a result, probiotics have become an excellent intervention tool for the spread of pathogenic bacteria and bacteria resistant to antibiotics [163]. In practice, the starting point was muscle mass gain due to the pressure to improve bird mass gain. A breakthrough in this type of therapy required an understanding of the extremely complex ecosystem in the intestines [164]. Currently, breeders are focusing on controlling and preventing the spread of pathogenic microorganisms. In the future, the feed additive industry may focus on the benefits, such as the maintenance of normal microflora or more precise selection of strains and doses, but this may be a more difficult process as the form of the disease changes with the animal husbandry conditions. Future research on probiotics should investigate the interaction between specific bacterial strains. Metabolites produced by interactions may have toxic effects. This is what happens, for example, with *Clostridium perfringens*, which kills even closely related strains [165]. However, bacteria may live in symbiosis and the metabolites of one strain may have a positive effect on another type of bacteria, creating a source of nutrients. This is a cross-feeding phenomenon which will be an important reference point for creating balance in the digestive system of birds. To maximize the effects of probiotics, they should not be treated as simple alternatives to antibiotics. The important correlation among host, feed, and microorganisms must be taken into account [166].

## 10. Conclusions

This review on using probiotics as feed additives confirms that they are worthy successors to AGP, reproducing their positive effects on the raw material obtained. Meat parameters, such as cholesterol, fatty acid profile, and oxidative stability of the meat are all improved by the addition of probiotics. Egg shells are less susceptible to injury, the weight of eggs, and their size is increased, as well as the laying itself. The use of probiotics also has a therapeutic basis, due to its bactericidal and bacteriostatic properties. This is due to competitive interactions between probiotic and pathogenic bacteria and innate immune stimulation. Probiotics help to prevent campylobacteriosis, coliobacteriosis, and salmonellosis, which are among the leading zoonoses. In addition to probiotic immunomodulation, these feed additives leave a positive impact on the gastrointestinal morphology itself, especially the villi and intestinal cups of the intestines. During heat stress, the overall homeostasis of the body is disturbed, leading to abnormal metabolism caused by of abnormal levels of thyroid hormones. The effects are visible at the molecular level by changing the expression of genes, such as heat shock proteins. Despite their versatility and advantages, probiotics do have a few disadvantages. Research on the effects of probiotics in rooster reproduction flocks showed a high concentration of probiotic bacteria in the cloaca. This leads to reduced sperm quality and may be the cause of reduced fertilization and even infertility, which is a huge concern to poultry producers. Probiotics are commonly known to be sensitive to temperature and humidity, which may impact their effectiveness. The type of dehydrator and storage conditions affect the structure of the bacteria, therefore it is important to be familiar with the production processes. When comparing probiotics with other feed additives, probiotics can most often be found in combination with other additives. This demonstrates the great versatility and compatibility of probiotics.

Future prospects for probiotics are promising as preparations with more strains are increasingly being studied, as well as interactions between them, such as cross-feeding. Correlations among host, feed, and microorganisms are also being studied. Better knowledge of these additives and more precise selection of their composition and recommended dose will improve their benefits in poultry production.

## Figures and Tables

**Figure 1 animals-11-01620-f001:**
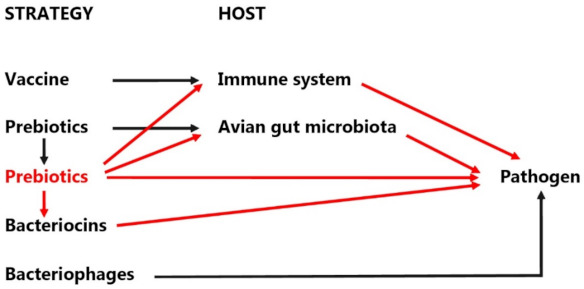
Potential pathways of the strategies in progress to reduce avian gut pathogens. Red arrows represent probiotic pathways [14].

**Figure 2 animals-11-01620-f002:**
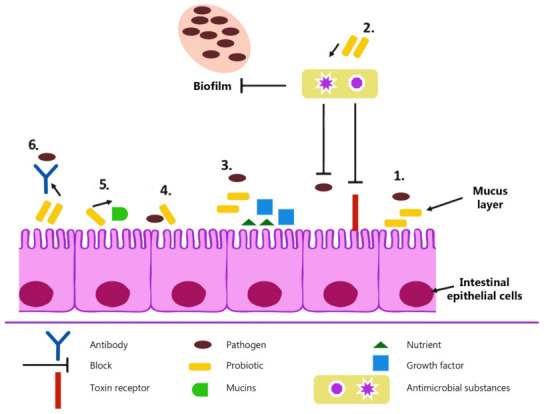
The possible mechanisms of probiotic action. (**1**) Competitive exclusion of pathogenic microorganisms. (**2**) Production of antimicrobial substances. (**3**) Competition for growth factors and nutrients. (**4**) Enhancement of adhesion to intestinal mucosa. (**5**) Improvement of epithelial barrier function. (**6**) Improvement of secretion of IgA [86].

**Figure 3 animals-11-01620-f003:**
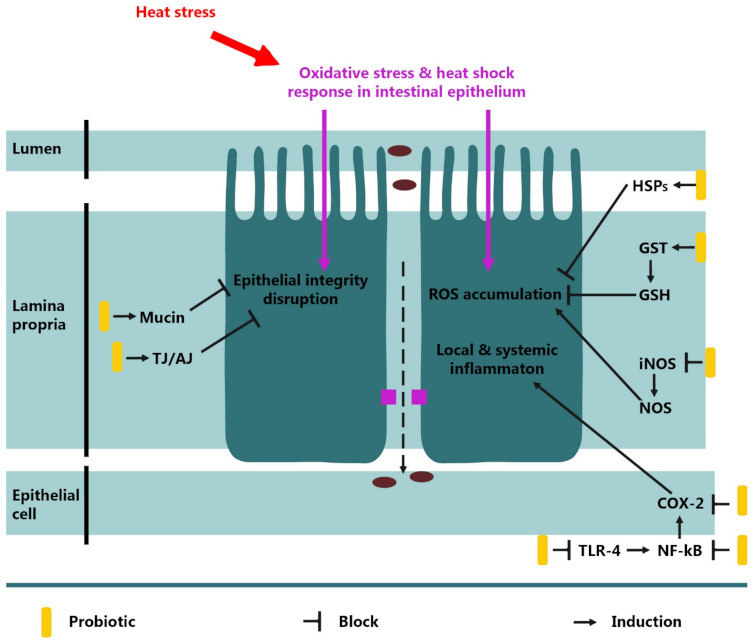
The damaging effect of heat stress on intestinal barrier and the pathways for the protective effects of probiotics [86].

**Table 1 animals-11-01620-t001:** Examples of probiotic preparations and strains used [119].

Name of the Preparation	Bacterial Strain	Species	Concentration
Acid-Pak-4-Way		*Lactobacillus acidophilus*	
		*Enterococcus faecium*	
Biogen D		*Bifidobacterium bifidum*	
		*Lactobacillus acidophilus*	
		*Pediococcus faecium*	
Lactiferm	M-74	*Enterococcus faecium*	
Oralin	DSM 10663/NCIMB10415	*Enterococcus faecium*	
Probiomix		*Bifidobacterium bifidum*	
		*Lactobacillus amylovorus*	
		*Enterococcus faecium*	
Probios		*Lactobacillus acidophilus*	
		*Lactobacillus casei*	
		*Lactobacillus plantarum*	
		*Enterococcus faecium*	
GutCare^®^ PY1	DSM 32315	*Bacillus subtilis*	4 × 106 CFU/mL

**Table 2 animals-11-01620-t002:** Comparison of the effects of various feed additives on the development of internal organs in poultry [126].

Dietary Treatment				
Factor	Control	Prebiotic	Probiotic	Synbiotic
Proventriculus	8.42	8.51	9.05	8.40
Gizzard	43.12	46.15	43.07	44.02
Liver	64.77	66.65	61.72	62.15
Spleen	1.87	1.98	2.11	2.06
Bursa	2.28	2.14	2.36	2.25

**Table 3 animals-11-01620-t003:** Strategies in progress to control *Campylobacter* spp. at the farm level [14].

Strategy	Principle	Advantage	Drawback	Way of Administration
Vaccination	Improvement of the immune response against *Campylobacter* spp.	Easy to use	Antigenic variability of *Campylobacter* spp. strains	Subcutaneously Oral
Bacteriophage therapy	Use of specific bacterial virus to kill *Campylobacter* spp.	Rapid action	Selection of resistant *Campylobacter* spp. strains Production cost Diversity of *Campylobacter* spp. strains	Subcutaneously Oral
Bacteriocin treatment	Use of bacteria-produced antimicrobial compounds against *Campylobacter* spp.	Easy to use	Production cost Variable sensitivity of *Campylobacter* spp. strains	Oral
Prebiotics	Incorporation of feed additives to improve beneficial avian gut microbiota	Easy to use Production cost	Dependence on the avian gut microbiota	Oral
Probiotics	Administration of beneficial microorganisms with anti-*Campylobacter* spp. activity	Easy to produce and to use Production cost Mix of multiple species Different ways of inhibiting *Campylobacter* spp.	Variable sensitivity of *Campylobacter* spp. strains	Oral

## Data Availability

Not applicable.

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
