# Peer review of "Overview of the Use of Probiotics in Poultry Production"

_animals, 2021, doi:10.3390/ani11061620_

Round 1

Reviewer 1 Report

This article provides an overview of research findings on the effects of probiotics in poultry production. Individual chapters/subchapters contain information on the impact of probiotics on production results (BW, BWG, FCR, survival), carcass composition and meat quality, intestinal microbiota composition, histological features of the small intestine (mainly jejunum), development of internal organs, bone condition, immune system, etc. Before publishing in Animals, the paper requires additions and corrections. The list of recommend changes is given below:

General comments

References must have „abbreviated name journal”

Please check the compliance of the cited article in the text with the items in References. There are mistakes.

Detailed comments

L4-5, please provide for individual authors the initials of the name and surname used in the Author Contributions section, add e-mail of each authors

L50 add information about the forecast of growth in the production of meat, fruit and vegetables until 2050

L63 in Poland, EU or in the world are 30 probiotic preparations registered, are they all multi-component probiotics - preparations?

L90 a space before "[10]"

L109 according to Refrences must be "Yang et al." instead of "Yang"

L146 according to Refrences must be "Keener et al." instead of "K.M. Keener "

L181-182 only "AGPs" or antibiotcs?

L223 according to References "44" is Hossain et al. not Abd-El-Hack

L242 according to References "Khempaka et al." is item 54, not 55

L 259 antibacterial and antifungal [60] instead of the current form

L338 space before Figure 2

L343 400x or 400X?

L371 Figure 3 instead of "figure 3"

L431 "0.175 g" for "0. 175g "

L456 S (less than 48 g) instead of the S

L457 L (73 g to above 63 g ) instead of L

L491 flocks instead of herds

L514 According to References chapter 121 is "Behrouz et al." not "Popova", please correct References

L518 Table 2 not Table 1, please change the table title. There are data for probiotics, synbiotic prebiotics, acidifier, please remove "P values", data from two articles, so where is this                        P value?

 L551 by References the RASFF report is 134 not 133

L637 T3 (triiodothyronine) and T4 (thyroxine) instead of the current form

L719 „flocks” instead of herds

L746-1175 in Reference chapter must be abbreviated name journal

L874 delete title in Polish

L878 check the conference record according to the authors' instructions for Animals

L896-898 article title, please use uppercase and lowercase letters like other items in References chapter, not printed

L944, 968, 1010, 1027, authors' names, please use uppercase and lowercase letters as other items in the References chapter, not printed

L980-982 article title, please use uppercase and lowercase letters as other items in References chapter, not printed

1087 and 1115 Please check the correctness of the citation

Author Response

Response to Reviewer 1 Comments
Point 1: References must have „abbreviated name journal”
Response 1: Names of the journals are shorter now.
Point 2: Please check the compliance of the cited article in the text with the items in References. There are mistakes.
Response 2: The citations in the text correspond to the numbers in the references now.
Point 3: L4-5, please provide for individual authors the initials of the name and surname used in the Author Contributions section, add e-mail of each authors
Response 3: Information about initials and e-mails are added and correct now.
Point 4: L50 add information about the forecast of growth in the production of meat, fruit and vegetables until 2050
Response 4: Information on individual projected data for 2050 has been added, taking a general view as I to individual food production sectors.
Point 5: L63 in Poland, EU or in the world are 30 probiotic preparations registered, are they all multi-component probiotics - preparations?
Response 5: The information has been supplemented, it was about probiotics in the European Union (as requested by the Reviewer), multi-ingredient preparations are allowed.
Point 6: L90 a space before "[10]"
Response 6: Space has been added.
Point 7: L109 according to References must be "Yang et al." instead of "Yang"
Response 7: The author's entry in the text has been corrected.
Point 8: L146 according to References must be "Keener et al." instead of "K.M. Keener "
Response 8: The author's entry in the text has been corrected.
Point 9: L181-182 only "AGPs" or antibiotcs?
Response 9: Of course, it was about the AGP’s ban.
Point 10: L223 according to References "44" is Hossain et al. not Abd-El-Hack
Response 10: The author's entry in the text has been corrected.
Point 11: L242 according to References "Khempaka et al." is item 54, not 55
Response 11: Reference is correct now.
Point 12: L 259 antibacterial and antifungal [60] instead of the current form
Response 12: Duplicate citation has been removed from the text.
Point 13: L338 space before Figure 2
Response 13: Space has been added.
Point 14: L343 400x or 400X?
Response 14: Of course we write magnification "x" as lowercase, this has been corrected.
Point 15: L371 Figure 3 instead of "figure 3"
Response 15: Letter is correct now.
Point 16: L431 "0.175 g" for "0. 175g "
Response 16: Space has been corrected.
Point 17: L456 S (less than 48 g) instead of the S
Response 17: The naming has been corrected.
Point 18: L457 L (73 g to above 63 g ) instead of L
Response 18: The naming has been corrected.
Point 19: L491 flocks instead of herds
Response 19: The naming has been corrected.
Point 20: L514 According to References chapter 121 is "Behrouz et al." not "Popova", please correct References
Response 20: The author's entry in the text has been corrected.
Point 21: L518 Table 2 not Table 1, please change the table title. There are data for probiotics, synbiotic prebiotics, acidifier, please remove "P values", data from two articles, so where is this P value?
Response 21: The title of the table has been corrected. P values have been removed.
Point 22: L551 by References the RASFF report is 134 not 133
Response 22: Reference is correct now.
Point 23: L637 T3 (triiodothyronine) and T4 (thyroxine) instead of the current form
Response 23: The unfolding of the names of the hormones was added as recommended.
Point 24: L719 „flocks” instead of herds
Response 24: The invalid name has been corrected.
Point 25: L746-1175 in Reference chapter must be abbreviated name journal
Response 25: All journal names in references now are abberatived.
Point 26: L874 delete title in Polish
Response 26: The polish title has been deleted.
Point 27: L878 check the conference record according to the authors' instructions for Animals
Response 27: Citation has been corrected, it was missing informations abut country, location and the name of the conference.
Point 28: L896-898 article title, please use uppercase and lowercase letters like other items in References chapter, not printed
Response 28: The notation has been changed to correct.
Point 29: L944, 968, 1010, 1027, authors' names, please use uppercase and lowercase letters as other items in the References chapter, not printed
Response 29: The notation has been changed to correct.
Point 30: L980-982 article title, please use uppercase and lowercase letters as other items in References chapter, not printed
Response 30: The notation has been changed to correct.
Point 31: 1087 and 1115 Please check the correctness of the citation
Response 31: Both citations were corrected, there was an authorship error that was fixed.

Reviewer 2 Report

Thanks for your reply.

What is the role of synthetic AGPs in poultry production in terms of the action mechanism?  Many similar reviews have been published up to now, and the description is one of the points that distinguish from such papers, which also mean the academic significance. L76 is not enough for that. 

Author Response

Response to Reviewer 2 Comments

Point 1: What is the role of synthetic AGPs in poultry production in terms of the action mechanism?  Many similar reviews have been published up to now, and the description is one of the points that distinguish from such papers, which also mean the academic significance

Response 1: I created a separate mid-length chapter dedicated to AGP. There I mentioned doubts about their mechanisms of action as well as the effects of using these supplements. The turning point is studies showing conflicting information with studies that are old and I have tried to show that despite the ban on use, the topic of the mechanism of action of APG has not been fully exhausted. Below I present the study I have discussed:

AGPs and Synthetic Growth Promoters (SGPs) are substances that had their heyday many years ago. In subtherapeutic concentrations they influenced the improvement of production indices such as body weight, FCR or daily gains. Their spectrum of action also included antimicrobial mechanisms, mainly targeting Gram-positive bacteria [17]. Their use to improve animal performance and rapid growth has maximized animal production results, while their mechanisms of action in this direction are not fully understood. Recent related knowledge highlights the possibility of manipulation of the gut microflora; AGPs have been shown to alter the diversity of gut bacteria, including beneficial LABs [18–20]. In the context to LAB bacteria, depending on the substance this effect varies [21]. Mecha-nisms of AGP action reach also to modulation of the animal immune system affecting its modulation, however, it has been shown that these reactions are different depending on the substance used, for example avilamycin affects the inhibition of bacterial protein syn-thesis, which release smaller amounts of proinflammatory compounds [22]. Also the use of these feed additives has an effect on the amount of vitamins, nucleosides, amino acids or fatty acids metabolized, interestingly studies have shown an increase in their levels. In contrast, the most shocking information is the increase in Polyunsaturated fatty acids (PUFA) [23]. Contradictory information on the topic of AGPs requires further study, and variation may originate from environmental differences affecting the study of external conditions, individual animal microflora composition, or animal health status. AGPs have been withdrawn due to undeniable residues in animal products, water, and soil, with negative consequences in terms of antibiotic resistance and allergies [24]. Their ani-mal performance-enhancing and antimicrobial properties are undeniable, while their mechanisms of action need to be understood more and compared with the advantages and disadvantages of other alternative substances used in agriculture.

Point 2: L76 is not enough for that.

Response 2: I have added information about why probiotics are used. Actually, the information may have been coherent, but for a better reading experience, I added a brief information about it, on the mechanisms of action. Below I present the study I have discussed:

Probiotics have many advantages and few disadvantages. The prospect of using pro-biotics in poultry production is clearly positive. Prophylactic use of probiotics occurs through antagonistic actions on other microorganisms and in competition for adhesion receptors or nutrients needed for their survival and some mechanisms like intestinal epi-thelial function and status. They also affect animal health as well as production perfor-mance, which will be developed later [16].

This manuscript is a resubmission of an earlier submission. The following is a list of the peer review reports and author responses from that submission.

Round 1

Reviewer 1 Report

The MS reviews the use of probiotics in poultry production aiming, as stated in the title, to “compare probiotics to other feed additives as an alternative to AGP”.

Research on the use of probiotics in poultry production and, generally, in the zootechnical field has made great strides in the last decades end several reviews are available, as the authors surely know. However, while it is appreciable the authors attempt compare probiotics whit other bioactive dietary additives, the aim was not achieved, since the MS looks like a list of feed additives (before) and of probiotics advantages (after). No comparison is made along the different sub-chapters.

Moreover:

  • in the abstract (line 43) there is a sentence regarding the probiotics side effect on goat that is totally spun off the research field you are dealing with, as no other comparison whit mammals is discussed in the test.
  • the 2° subtitle (Probiotics and other feed additives) is not correct, since you are dealing (2.1) whit Phytobiotics which are PREbiotics (not living organisms) and of course, later, other additives; however different bioactive compounds, such as chitin and M/LCSFAs (which effects are studied in mammals, birds and fish) are not considered at all.
  • the MS contains 1 figure that alone could be avoided and a fig.2 which should be a scheme that plotted in this way does not provide any useful information
  • Conclusions are too long
  • the collection of information is somewhere without references and seems to be a summary of what has been read in other papers, whit a scarce processing footprint.
  • English is somewhere to be improved

Overall, since the depth of the reviews in this field has reached high levels in particular in the last years, I suggest major revision of the MS, which, aiming on the goal stated in the title, has to be improved whit a deeper elaboration of the information collected and enriched with schemes, tables and pictures in order to provide a new point of view in the fields and better to catch readers interest.

Reviewer 2 Report

In 2015 the value of the probiotic market reached $33.19 billion [6]. In 2020, 74 the value of the market is $46.55 billion [6]

  • I suggest to move the refernce numer 6 or insert a more recent reference.
  • Insert, if is possible, the Figure 1 with more resolution.
  • line 336 remove comma
  • the names of bacteria are write in italic, please change in the text.

Reviewer 3 Report

The paper contains an overview of current knowledge on the effectiveness of the use of probiotics and other feed additives as an alternative to Antibiotic Growth Promoters (AGP) in commercial poultry production. The article must be completed and corrected before publishing in Animals. The list of proposed changes is below.

L3 suggest "Antibiotic Growth Promoters in commercial poultry production" instead of AGP

L28 please add information on the ban of the use of AGP outside the EU, the composition of microorganisms usually used in probiotics, methods of administration of probiotics in poultry, the number of probiotic preparations registered as feed additives in the EU.

L63 is it about feed antibiotics (AGP), antibiotics are still used therapeutically and prophylactically in poultry production

L66 ban of the use of antibiotics or AGP?

L69 antibiotics or AGP, Goodvalley produces pork without antibiotics. Combined with those profiled and therapautically. For poultry, there are no such meat production systems yet.

L83: Improvement, but what? Quantity and quality of poultry raw materials

L92 Yang [13] instead of Yang [2009], correction of subsequent references needed

L124 delate K.M.

SDS (full name) instead of SDS

L155 antibiotics or AGP

L199 1010 and 1011 CFU/g is correct?

L254+ add information about the effect of probiotics on the composition of the carcass, proximate chemical composition, minerals, microstructure and texture of meat (4-5 new papers)

Please also add papers with a insignificant impact of probiotics on the features discussed and required research in the future.

What determines the positive impact of probiotics in one study and the lack in others. How do environmental conditions (bad, good) and genotype affect exposure affect probiotics on the characteristics studied?

Add a report on the impact of probiotics on the body's development, length, diameter, weigth of the intestine and its segments, and other internal organs, biometric and mechanical characteristics of bones

(4-5 new works)

L322 and how it relates to RASFF data - see EFSA reports

L452+ In my opinion, the Conclusions chapter should be shortened

Author Contributions, Funding, Acknowledgments, Conflict of Interest - must be done in accordance with the instructions for the authors of the Animals journal (https://www.mdpi.com/journal/animals/instructions)

References Titles in Polish, please translate into English, References no. 5 and 9

L734 „Poultry Science” instead of Poultry science

L750 „Reviewers” instead of reviewers

L757 2017 (in bold), 17, 591 instead of the current form.

Reviewer 4 Report

The manuscript reviewed probiotics compared to other feed additives as an alternative to AGP originally. Many information were described and summarized, however what AGPs are NOT explained. This definition is very important to evaluate which functions of additives are resembled to AGPs or unique. It is also concern if Fig. 2 is original or not, and the reviewer cannot feel the novelty for the figure.